# InsVP: Efficient Instance Visual Prompting from Image Itself

## ABSTRACT

Visual prompting is an efficient methodology for finetuning pretrained visual models by introducing a small number of learnable parameters while keeping the backbone frozen. However, most existing visual prompting methods learn a shared prompt for all samples, making it challenging to grasp distinct characteristics among diverse samples, thereby limiting the model's performance. While other methods partially address this issue through sample clustering and learning multiple prompts, they still struggle to capture nuanced differences among instances and incur significant parameter overhead. Therefore, to comprehensively and efficiently leverage discriminative characteristics of individual instances, we propose an **Ins**tance **V**isual **P**rompting method, called **InsVP**. Initially, the instance image prompt is introduced to extract both crucial and nuanced discriminative information from the original image itself and is overlaid onto the input image. Furthermore, the instance feature prompt is designed to capture both commonalities and characteristics among individual instances, fed into the model's intermediate layers to facilitate feature extraction. Consequently, the instance image and feature prompts complement each other, enhancing the adaptation ability of pretrained models to extract discriminative features from individual instances. Extensive experiments on various large-scale benchmarks show that our InsVP achieves superior performance exceeding the state-of-the-art methods at a lower parameter cost. Our code will be released.

## CCS CONCEPTS

• **Computing methodologies** → **Computer vision**.

## KEYWORDS

Visual Prompting, Prompt Learning, Parameter-efficient Fine-tuning

## 1 INTRODUCTION

Over the past years, the deep learning community has widely embraced the pretraining-finetuning paradigm, which has played a pivotal role in propelling the field of computer vision (CV) [14, 15, 21, 22]. However, with the explosive growth of model size and data scale, such a conventional paradigm suffers from unaffordable storage and computational overheads [23]. Therefore, the latest efforts [5, 16, 17, 53] have concentrated on *how to adapt the pretrained models to a particular downstream task efficiently*. Notably, the emergence of visual prompting technology [3, 20, 23] has taken the lead in addressing this challenge. By introducing a small number

*ACM MM, 2024, Melbourne, Australia*

© 2024 Copyright held by the owner/author(s). Publication rights licensed to ACM.
ACM ISBN 978-x-xxxx-xxxx-x/YY/MM
https://doi.org/10.1145/nnnnnnn.nnnnnnn

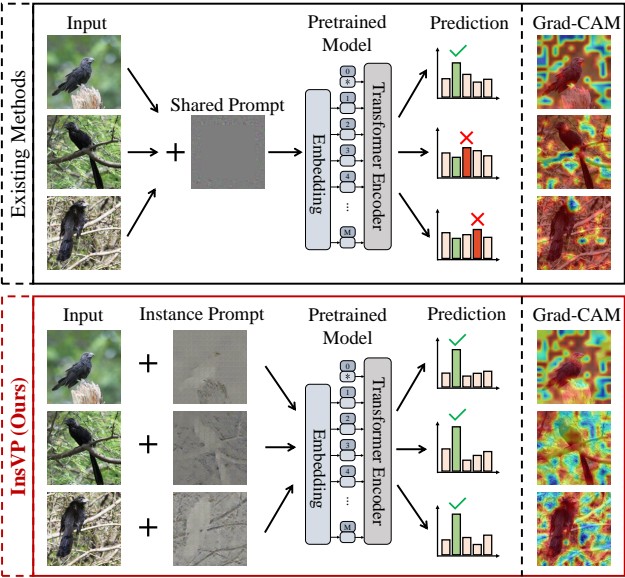

Figure 1: **Existing visual prompting methods [3, 16, 20, 23] train a shared prompt for all samples or within clusters, struggling to capture the distinct characteristics of individual instances. In contrast, our InsVP employs an image-driven instance prompt, capturing distinctive areas of instances and guiding the pretrained model to focus on them. The visualization results by Grad-CAM [39] verify that InsVP focuses on discriminative regions of images and achieves excellent performance.**

of learnable parameters, visual prompting can efficiently adapt the pretrained models to downstream tasks while keeping the whole pretrained backbone frozen.

Most existing visual prompting methods have opted for a shared prompt that is applied uniformly to all data [3, 16, 23, 48], disregarding the potentially significant variations among different data, as shown in Figure 1. Consequently, the learned prompts fail to capture the distinct characteristics of individual instances, considerably limiting the discriminative power of deep models [20]. To overcome this limitation, recent visual prompting methodologies [20] have endeavored to cluster samples and simultaneously learn a cluster-specific prompt for each cluster, thereby mitigating the issue to some extent. However, these approaches still struggle to capture the subtle nuances of individual instances, as even samples within the same cluster exhibit fine-grained distinctions. Moreover, these methods unavoidably introduce significant parameter overhead, substantially impeding the scalability of the pretrained model.

To address these limitations, we propose a novel and efficient visual prompting method **InsVP**, namely **Ins**tance **V**isual **P**rompting. As illustrated in Figure 1, InsVP extracts distinctive areas unique to

individual instances from the image itself, guiding the pretrained model to focus on the exclusive discriminative characteristics of the instances. To achieve this, we first design the image-level instance visual prompting to highlight the discriminative areas of instances in the input image, which comprises two complementary components of a patch prompt and a global prompt. By forwarding the input image to the designed lightweight prompters, the obtained patch prompt captures fine-grained local information from individual patches, meanwhile the global prompt gathers overall information from the entire image. Together, they complement each other in extracting the distinguishing regions of the instances, resulting in an instance image prompt that is overlaid onto the original image.

Furthermore, our InsVP also introduces the feature-level instance visual prompting to continuously incorporate instance information into the intermediate layers of the pretrained model. Considering both the commonalities and characteristics between different instances at the feature level, we design the learnable common prompt and the generated specific prompt respectively. Motivated by VPT [23], several learnable tokens are introduced as the common prompt to distill overarching patterns and fundamental attributes, fostering a comprehensive exploration of commonalities across all images. Moreover, a lightweight specific prompter is proposed to enhance the distinctive features specific to the individual instance from the input image itself. The collaboration between the common and specific prompts improves the adaptation capacity of the pretrained model to different instance samples, leading to superior performance. The main contributions of this work are:

- To address the limitations of existing visual prompting methods, we propose InsVP, an efficient approach aimed at comprehensively leveraging the discriminative instance-specific information of the input image itself to enhance the recognition capability of pretrained models.
- In InsVP, a novel image-level instance visual prompting scheme is designed to capture and emphasize the discriminative areas of different instances in the input image.
- Moreover, a complementary feature-level instance visual prompting model is developed in InsVP to direct the pretrained model to pay attention to the discriminative characteristics of the instances to facilitate feature extraction.
- Extensive experiments on various datasets show that our InsVP significantly outperforms the existing visual prompting methods with a much lower parameter cost.

## 2 RELATED WORK

### 2.1 Parameter-Efficient Finetuning

Vision Transformer (ViT) has made remarkable achievements in the field of computer vision [1, 7, 13, 30, 46]. However, with the rapid increase in model size, fully finetuning the pretrained ViT models for downstream tasks inevitably brings large storage and computing overhead. Therefore, recent works [16, 23, 53] started to focus on reducing the number of learnable parameters for efficient finetuning of pretrained models which can be broadly categorized into *partial tuning-based*, *extra module-based*, and *prompt learning-based* ones.

Partial tuning-based approaches [17, 35, 49, 54] aim to freeze the majority of the pretrained backbone while finetuning a small portion of the model parameters. For instance, such methods might only adjust the Linear/MLP heads [17, 21], or refine a part of layers within the backbone [35, 49, 54]. While these approaches are straightforward and simple to implement, they commonly exhibit a substantial performance gap when compared to fully finetuning [10, 32]. In contrast, extra module-based methods [5, 36, 38, 51, 53] design additional learnable plug-in architectures to finetune the pretrained model. [53] introduced an extra learnable side network while maintaining the original model frozen. Similarly, other studies [8, 36, 38] proposed to insert extra learnable residual units into the backbone. A limitation of these approaches lies in their customized nature for specific architectures, hindering generalizability to other models. Moreover, these modules obviously introduce more learnable parameters compared to partial tuning-based methods, making them difficult to apply in practice [16, 23].

### 2.2 Prompt Learning

Prompt learning techniques initially emerged in the field of natural language processing (NLP), involving the integration of a small set of learnable soft-prompt into input texts to tailor language models for specific downstream tasks [26–28]. Recent studies have extended prompt learning to visual tasks, termed visual prompt learning or visual prompting [3, 6, 16, 20, 23, 29, 48]. Compared with partial tuning-based and extra module-based methods, the visual prompting-based approaches introduce significantly fewer additional parameters and achieve better compatibility with models of different structures [16, 23].

Specifically, existing visual prompting methods usually follow two popular manners, task-level visual prompting [3, 16, 23, 41, 44, 48] and cluster-level visual prompting [20]. The former involves all downstream data samples learning a set of shared image prompts. VP [3] learned a single prompt for all samples, which is added around the image in the form of padding and then fed into the pretrained model together with the original image. VPT [23] introduced several learnable tokens as the prompt shared by all samples which are used in the multi-head self-attention (MSA) blocks of the pretrained ViT model. On the basis of VPT, $E^2$VPT [16] pruned the learned prompts from both the token-wise and segment-wise perspectives, which reduces the impact of unfavorable prompts while reducing the number of additional parameters required. However, the above methods fail to capture the discriminative characteristics of instances, limiting the model performance [20]. The latest cluster-level visual prompting method DAM-VP [20] involved clustering samples and learning a set of visual prompts for each cluster. This approach partly mitigates the above problem, yet still struggles to capture subtle differences among instances within the same cluster. In addition, it will bring large parameter overhead, resulting in poor scalability. In contrast, our proposed InsVP introduces lightweight prompters to generate an instance prompt for each sample.

## 3 INSVP: INSTANCE VISUAL PROMPTING

In this section, we illustrate the proposed visual prompting method *InsVP* in detail. InsVP aims to generate instance-specific visual prompts for each individual image to adapt the pretrained model

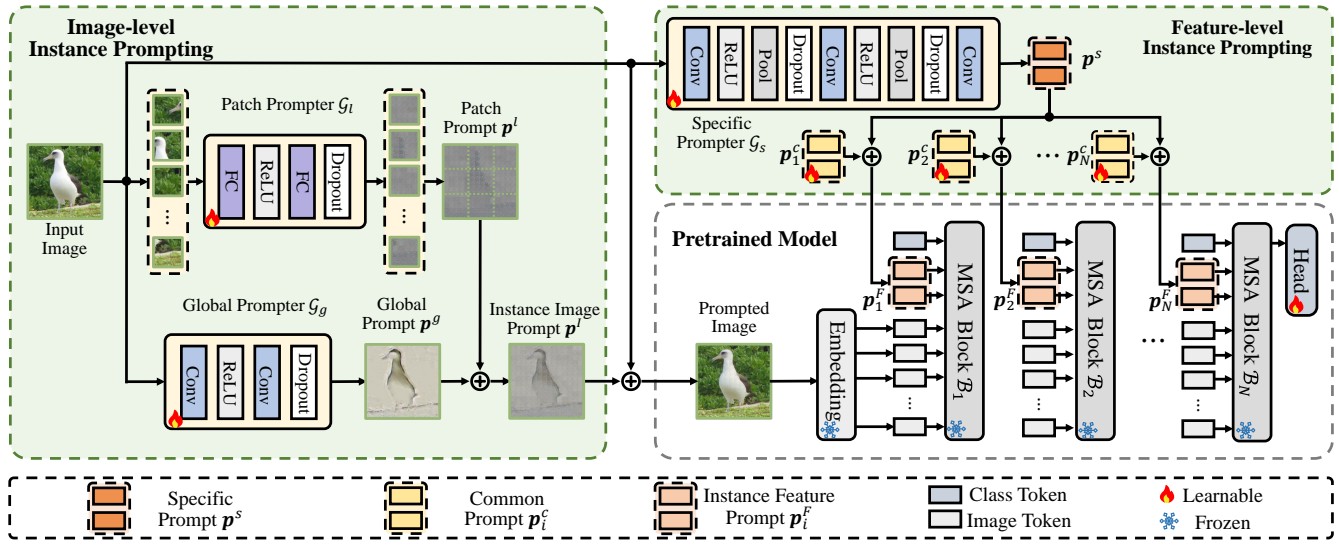

**Figure 2: The pipeline of our proposed InsVP. For each image instance, InsVP first utilizes two lightweight prompters $\mathcal{G}_l$ and $\mathcal{G}_g$ to generate its patch prompt $p^l$ and global prompt $p^g$ respectively. Then they are merged into the instance image prompt $p^I$, which is further superimposed on the original image as the input of the pretrained model. Furthermore, the specific prompter $\mathcal{G}_s$ is used to get the specific prompt $p^s$. It is merged with the learnable common prompt $p_i^c$ to form the instance feature prompt $p_i^F$, which serves as the input tokens of the MSA blocks.**

efficiently. The notations used in this paper are introduced in Section 3.1, followed by the introduction of the image-level instance prompting in Section 3.2 and feature-level instance prompting in Section 3.3. The overall pipeline of InsVP is depicted in Figure 2.

## 3.1 Preliminaries

For a pretrained Vision Transformer (ViT) [13] backbone $\mathcal{M}$, it contains $N$ MSA blocks $\{\mathcal{B}_j\}_{j=1}^N$ where each $\mathcal{B}_j$ consists of multi-head self-attention and feed-forward networks together with LayerNorm [2] and residual connections [18]. For an input image $x \in \mathbb{R}^{H \times W \times C}$, it is initially divided into several equally sized patches $\{x_i\}_{i=1}^M \in \mathbb{R}^{h \times w \times C}$, where $(H, W)$ is the size of image $x$, $C$ is the number of channels of $x$, $(h, w)$ is the size of patch $x_i$, $M$ is the number of patches. Each patch $x_i$ is then first embedded into a $d$-dimensional latent space as:

$$h_i^1 = \mathcal{E}(x_i),\qquad(1)$$

where $h_i^1 \in \mathbb{R}^d$, $\mathcal{E}(\cdot)$ donates the embedding layer of the backbone $\mathcal{M}$. Subsequently, all image tokens $\{h_i^1\}_{i=1}^M$ along with an additional classification token $c_1 \in \mathbb{R}^d$ are fed into the $N$ MSA blocks $\{\mathcal{B}_j\}_{j=1}^N$ to extract features as:

$$\left[c_{j+1}, h_1^{j+1}, \cdots, h_M^{j+1}\right] = \mathcal{B}_j\left(\left[c_j, h_1^j, \cdots, h_M^j\right]\right),\qquad(2)$$

where "[ ]" indicates stacking and concatenating on the sequence length dimension. Finally, the output $c_{N+1}$ from the last MSA block is passed through a classification head $\mathcal{H}$ to derive the predicted probability distribution $y$:

$$y = \mathcal{H}(c_{N+1})\qquad(3)$$

## 3.2 Image-level Instance Visual Prompting

To capture distinctive information of the input instance image $x$, we initially propose image-level instance visual prompting, which involves generating the instance image prompt for each input image to improve the performance of the pretrained model $\mathcal{M}$.

Specifically, two lightweight networks, the global prompter $\mathcal{G}_g$ and the patch prompter $\mathcal{G}_l$, are designed to readily explore the global and local discriminative characteristics of $x$. The global prompter $\mathcal{G}_g$ is composed of two layers of dilated convolution [50], along with a ReLU activation layer and a Dropout layer. The utilization of dilated convolution increases the receptive field of the global prompter $\mathcal{G}_g$ without adding extra parameters, providing it with a broader global perspective across the entire image. Building on this design, the global promotor $\mathcal{G}_g$ is capable of extracting global discriminative information from the original image $x$ via the generated prompt $p^g \in \mathbb{R}^{H \times W \times C}$, such as the object's position, shape, and contour details:

$$p^g = \mathcal{G}_g(x).\qquad(4)$$

Additionally, to fully capture the locally fine-grained information of the individual image patch $\{x_i\}_{i=1}^M$, a patch image prompter $\mathcal{G}_l$ is leveraged. To generate the patch prompt efficiently, the patch prompter $\mathcal{G}_l$ consists of two fully connected layers, taking a single image patch $x_i$ as input and producing the corresponding patch prompt $p_i^l$. This patch-based design significantly reduces the input and output dimensions of the fully connected layers, thereby decreasing the additional parameter overhead. By dividing the entire image $x$ into $M$ patches $\{x_i\}_{i=1}^M$, each patch $x_i$ is fed into the patch

prompter $\mathcal{G}_l$ to obtain $\boldsymbol{p}_i^l$ as below:

$$\boldsymbol{p}_i^l = \mathcal{G}_l\left(\boldsymbol{x}_i\right), \tag{5}$$

where $\boldsymbol{p}_i^l \in \mathbb{R}^{h \times w \times C}$. Subsequently, all $\{\boldsymbol{p}_i^l\}_{i=1}^M$ are concatenated in accordance with the relative order of the image patches $\{\boldsymbol{x}_i\}_{i=1}^M$, forming a complete patch prompt $\boldsymbol{p}^l \in \mathbb{R}^{H \times W \times C}$ for the input image $\boldsymbol{x}$.

Consequently, after generating the global image prompt $\boldsymbol{p}^g$ and the patch prompt $\boldsymbol{p}^l$, the instance image prompt $\boldsymbol{p}^I \in \mathbb{R}^{H \times W \times C}$ for the input image $\boldsymbol{x}$ is obtained by directly merging both two prompts in a linear combination manner to amalgamate the global and local information from the image itself:

$$\boldsymbol{p}^I = \beta_I \cdot \boldsymbol{p}^g + (1 - \beta_I) \cdot \boldsymbol{p}^l, \tag{6}$$

where $\beta_I$ is a pre-defined weight hyper-parameter. Finally, the image prompt $\boldsymbol{p}^I$ is superimposed onto the original image, serving as the input $\tilde{\boldsymbol{x}}$ of the pretrained model $\mathcal{M}$:

$$\tilde{\boldsymbol{x}} = \boldsymbol{x} + \boldsymbol{p}^I, \tag{7}$$

where $\tilde{\boldsymbol{x}} \in \mathbb{R}^{H \times W \times C}$ is the prompted image.

### 3.3 Feature-level Instance Visual Prompting

To further enhance the adaptation ability of the pretrained model $\mathcal{M}$, we design feature-level instance visual prompting to continuously incorporate instance-specific information into the MSA blocks. Given that at the feature level, different instances may exhibit both similarities and differences in their characteristics, our feature-level instance visual prompting proposes to utilize a common prompt $\boldsymbol{p}^c$ and a specific prompt $\boldsymbol{p}^s$ to simultaneously capture the commonality information among different instances and the distinctive information of each individual instance respectively.

Motivated by VPT [23], we introduce a number of $L_p$ learnable tokens as the common prompt. For the sake of simplicity, we collectively denote all introduced tokens as $\boldsymbol{p}^c$ without further distinction. The common prompt $\boldsymbol{p}^c$ consists of distinct prompts added to each MSA block which is commonly used for all instances:

$$\boldsymbol{p}^c = \left\{\boldsymbol{p}_j^c\right\}_{j=1}^N, \tag{8}$$

where $\boldsymbol{p}_j^c \in \mathbb{R}^{d \times L_p}$ is the common prompt for $j$-th MSA block $\mathcal{B}_j$.

As for the specific prompt $\boldsymbol{p}^s \in \mathbb{R}^{d \times L_p}$, it is directly obtained from the image $\boldsymbol{x}$ itself. In detail, a three-layer convolutional network is designed as the specific prompter $\mathcal{G}_s$ to generate $\boldsymbol{p}^s$ for all MSA blocks:

$$\boldsymbol{p}^s = \mathcal{G}_s\left(\boldsymbol{x}\right). \tag{9}$$

Specifically, the specific prompter $\mathcal{G}_s$ gradually employs convolutional and pooling layers to encode the image $\boldsymbol{x}$ into $\boldsymbol{p}^s \in \mathbb{R}^{h \times w \times (C \cdot L_p)}$. Finally, the output $\boldsymbol{p}^s$ is reshaped into the specific prompt $\boldsymbol{p}^s \in \mathbb{R}^{d \times L_p}$.

Subsequently, the complete instance feature prompt $\boldsymbol{p}_j^F$ is formed by readily adding up the common prompt $\boldsymbol{p}_j^c$ and specific prompt $\boldsymbol{p}^s$:

$$\boldsymbol{p}_j^F = \beta_F \cdot \boldsymbol{p}_j^c + (1 - \beta_F) \cdot \boldsymbol{p}^s, \tag{10}$$

where $\beta_F$ is a pre-defined weight hyper-parameter. The instance feature prompt $\boldsymbol{p}_j^F$ is combined with the image patch tokens $\{\boldsymbol{h}_i^j\}_{i=1}^M$

and the classification token $\boldsymbol{c}_j$, then collectively fed into the MSA block for feature extraction:

$$\left[\boldsymbol{c}_{j+1}, \hat{\boldsymbol{p}}_{j+1}^F, H^{j+1}\right] = \mathcal{B}_j\left(\left[\boldsymbol{c}_j, \boldsymbol{p}_j^F, H^j\right]\right), \tag{11}$$

where $H^j = \left[\boldsymbol{h}_1^j, \boldsymbol{h}_2^j, \cdots, \boldsymbol{h}_M^j\right]$. Notably $\hat{\boldsymbol{p}}_{j+1}^F$ get from $\mathcal{B}_j$ is not utilized in the next block $\mathcal{B}_{j+1}$. Extensively extracting distinctive characteristics from images, the instance feature prompt $\boldsymbol{p}^F$ and image prompt $\boldsymbol{p}^I$ facilitate the pretrained model in capturing discriminative features of individual instances. Ultimately, the $\boldsymbol{c}_{N+1}$ obtained from the final MSA block $\mathcal{B}_N$ undergoes processing via a classification head $\mathcal{H}$ to get the predicted probability distribution $\boldsymbol{y}$ via Equation 3.

### 3.4 Overall Optimization

As mentioned above, our InsVP introduces only a few additional parameters:

$$\boldsymbol{G} = \left\{\mathcal{G}_p, \mathcal{G}_g, \mathcal{G}_s, \boldsymbol{p}^c\right\}. \tag{12}$$

The extra parameters of InsVP are notably lightweight compared to the pretrained model and other visual prompting methods [16, 20, 23] as demonstrated in Section 4.4.5. Following previous works [16, 20, 23], during training, we maintain the pretrained model's encoder frozen while allowing only the classification head to be trainable. The optimization objective is as follows:

$$\arg\min_{\mathcal{G}, \mathcal{H}} \mathcal{L}_{ce}\left(\boldsymbol{y}, y_{gt}\right), \tag{13}$$

where $\mathcal{L}_{ce}$ is cross-entropy loss, $y_{gt}$ is the label of image $\boldsymbol{x}$.

## 4 EXPERIMENTS

### 4.1 Datasets

Building upon previous works [16, 20, 23], the experiments are conducted on four fine-grained datasets: CUB-200-2011 [43], NABirds [19], Oxford Flowers [34], and Stanford Dogs [24]. Additionally, following DAM-VP [20], we also perform experiments on another six commonly used visual datasets, including DTD [11], Food101 [4], Cifar100 [25], Cifar10 [25], GTSRB [40], and SVHN [33]. Following [23], for datasets with only publicly available train and test sets, we randomly split the train set into a train set (90%) and a validation set (10%), and utilize the validation set to determine hyper-parameters.

Additionally, we conduct experiments on the VTAB-1k benchmark [52] following [16, 23]. VTAB-1k is a benchmark that tests how well visual models perform across 19 different tasks. These tasks fall into three categories: Natural, for everyday image recognition; Specialized, for specific areas like medical images; and Structured, for understanding complex scenes, such as 3D object recognition.

### 4.2 Comparison Methods

We compare our InsVP with both parameter-efficient finetuning methods and visual prompting methods. We also report the fully-tuning results as a baseline. For parameter-efficient finetuning, we report the results of partial tuning-based methods including linear probing [21], Partial [49], MLP [17], and the results of extra module-based methods including Sidetune [38], Bias [53], Adapter [5],

**Table 1: The comparison results against state-of-the-art methods on ten datasets. *Partial*, *Extra*, and *Prompting* represent partial tuning-based, extra module-based, and prompt learning-based parameter-efficient finetuning methods respectively. Following their paper, ILM-VP, Yoo et al and AutoVP utilize ResNeXt-101-32x8d [47], MoCo v3 trained ViT-B/16 and CLIP [37] as the backbone respectively. The best results are bolded and the second-best results are underlined.**

| | Methods | Publication | DTD | CUB | Birds | Dogs | Flowers | Food | Cifar100 | Cifar10 | GTSRB | SVHN | Avg |
|---|---|---|---|---|---|---|---|---|---|---|---|---|---|
| | Full [21] | CVPR 2022 | 64.3 | 87.3 | 82.7 | 89.4 | 98.8 | 84.9 | 68.9 | 97.4 | 97.1 | 87.4 | 85.8 |
| *Partial* | Linear [21] | CVPR 2022 | 63.2 | 85.3 | 75.9 | 86.2 | 97.9 | 84.4 | 63.4 | 96.3 | 68.0 | 36.6 | 75.7 |
| | Partial-1 [49] | NeurIPS 2014 | 70.1 | 85.6 | 77.8 | 85.5 | 98.2 | 83.8 | 78.0 | 95.0 | 89.3 | 82.4 | 84.6 |
| | MLP-3 [9] | CVPR 2020 | 66.2 | 85.1 | 77.3 | 84.9 | 97.9 | 84.6 | 77.5 | 93.2 | 71.8 | 60.5 | 79.9 |
| *Extra* | Bias [38] | NeurIPS 2017 | 69.8 | 88.4 | 84.2 | 91.2 | 98.8 | 86.2 | 82.9 | 96.9 | 89.9 | 82.5 | 87.1 |
| | Sidetune [53] | ECCV 2020 | 57.7 | 84.7 | 75.8 | 85.8 | 96.9 | 78.7 | 68.8 | 90.4 | 90.9 | 80.5 | 81.0 |
| | Adapter [5] | NeurIPS 2020 | 62.7 | 87.1 | 84.3 | 89.8 | 98.5 | 86.0 | 74.2 | 97.7 | 91.1 | 36.3 | 80.8 |
| | AdaptFormer [8] | NeurIPS 2022 | 64.2 | 87.3 | 84.1 | 88.1 | 98.4 | 85.7 | 79.4 | 96.5 | 91.7 | 83.0 | 85.8 |
| *Prompting* | VP [3] | arXiv 2022 | 59.5 | 84.6 | 77.7 | 84.5 | 97.7 | 80.5 | 78.7 | 94.2 | 89.4 | 87.6 | 83.4 |
| | VPT [23] | CVPR 2022 | 65.8 | 88.5 | 84.2 | 90.2 | 99.0 | 83.3 | 78.8 | 96.8 | 90.7 | 78.1 | 85.5 |
| | DAM-VP [20] | CVPR 2023 | 73.1 | 87.5 | 82.1 | 92.3 | **99.2** | 86.9 | 88.1 | 97.3 | 90.6 | 87.9 | 88.5 |
| | ILM-VP [6] | CVPR 2023 | 41.4 | 7.7 | 11.6 | 87.6 | 27.9 | 23.0 | 45.9 | 81.7 | 59.9 | 81.4 | 46.8 |
| | Yoo et al [48] | ICML 2023 | 69.6 | 82.9 | 76.0 | 83.4 | 93.7 | 82.9 | 85.8 | 97.3 | 92.6 | 90.1 | 85.4 |
| | E²VPT [16] | ICCV 2023 | 66.8 | 88.4 | 84.2 | 91.3 | 99.0 | 84.0 | 80.4 | 97.1 | 91.0 | 79.2 | 86.1 |
| | TransHP [45] | NeurIPS 2023 | 68.4 | 87.1 | 82.7 | 91.5 | 98.6 | 85.5 | 86.9 | 97.3 | 91.3 | 82.9 | 87.2 |
| | LION [44] | AAAI 2024 | - | - | - | 83.6 | 90.5 | - | 65.4 | 90.8 | - | - | - |
| | AutoVP [41] | ICLR 2024 | 62.5 | 85.4 | 83.5 | 90.3 | 90.4 | 82.3 | 77.9 | 95.2 | 93.1 | 92.9 | 85.4 |
| | **InsVP(Ours)** | **This Paper** | **74.5** | **89.3** | **84.6** | **93.6** | **99.2** | **89.5** | **91.3** | **98.4** | **96.1** | **96.1** | **91.3** |

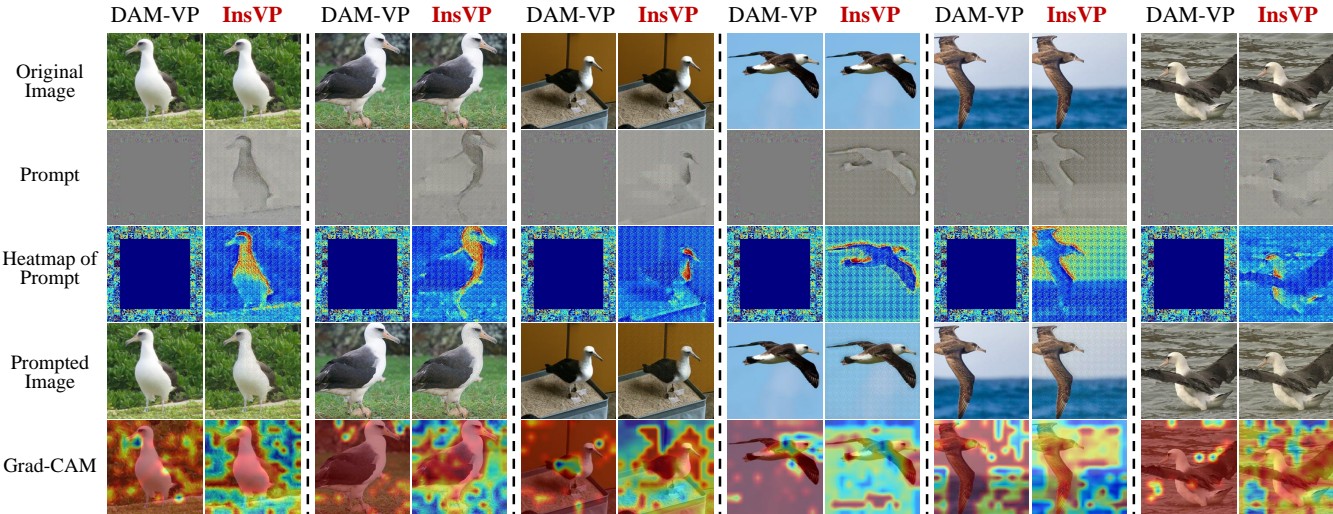

**Figure 3: Visualization results of various instance samples in CUB. We present the original images along with the prompts of DAM-VP and our InsVP for the instances. Moreover, the heatmaps of the prompts, the prompted image, and the corresponding Grad-CAM visualization results are also presented.**

AdaptFormer [8]. For visual prompting methods, we compare with the task-level visual prompting methods such as VP [3], VPT [23], ILM-VP [6], Yoo et al [48], E2VPT [16], LION [44], AutoVP [41] and the latest cluster-level visual prompting approach DAM-VP [20] and TransHP [45].

## 4.3 Implementation Details

Our experiments involve three pretrained vision models including the ViT-B/16 [13] and Swin Transformer [30] which are supervised by ImageNet-21k [12], and another ViT-B/16 that is learned via MoCo v3 [10]. Following DAM-VP [20], we train for 100 epochs on all datasets. We utilize the AdamW [31] optimizer for optimization and implement cosine annealing. The hyper-parameters length of instance feature prompt $L_p$, fusion weight of image prompts $\beta_I$, and fusion weight of feature prompt $\beta_F$ are set to 9, 0.7, and 0.5 respectively. The learning rate and weight decay on each dataset are detailed in the Supplementary Materials.

**Table 2: The comparison results against state-of-the-art methods on VATB-1k benchmark [52]. Following their paper, Yoo et al utilizes MoCo v3 trained ViT-B/16 as the backbone. Other methods utilize the ViT-B/16 [13] pretrained with supervised training on ImageNet-21k [12] as the backbone.**

| | Methods | VTAB-1k | | |
|---|---|---|---|---|
| | | Natural | Specialized | Structured |
| | Full [21] | 75.9 | 83.4 | 47.6 |
| Partial | Linear [21] | 68.9 | 77.2 | 26.8 |
| | Partial-1 [49] | 69.4 | 78.5 | 34.2 |
| | MLP-3 [9] | 67.8 | 72.8 | 30.6 |
| Extra | Bias [38] | 73.3 | 78.3 | 44.1 |
| | Sidetune [53] | 58.2 | 68.1 | 23.4 |
| | Adapter [5] | 70.4 | 77.1 | 33.4 |
| Prompting | VPT [23] | 78.5 | 82.4 | 55.0 |
| | Yoo et al [48] | 74.8 | 83.4 | 49.1 |
| | E$^2$VPT [16] | 80.0 | 84.4 | 57.4 |
| | **InsVP(Ours)** | **81.8** | **85.2** | **58.4** |

**Table 3: The comparison results of visual prompting methods on different network architectures. The Swin Transformer pretrained on ImageNet-21k is utilized as the backbone.**

| Methods | CUB | Birds | Cifar100 | GTSRB | SVHN |
|---|---|---|---|---|---|
| Full [21] | 89.7 | 86.8 | 73.3 | 97.1 | 91.2 |
| VP [3] | 86.5 | 82.9 | 80.6 | 82.4 | 80.3 |
| VPT [23] | 90.0 | 85.4 | 80.5 | 86.2 | 87.8 |
| DAM-VP [20] | 90.4 | 86.9 | 88.1 | 86.8 | 81.7 |
| E$^2$VPT [16] | 90.3 | 85.2 | 83.5 | 87.1 | 88.2 |
| **InsVP** | **91.2** | **87.7** | **90.3** | **90.1** | **91.4** |

**Table 4: The comparison results of visual prompting methods on different pretraining methods. The MoCo-v3 learned ViT-B/16 is utilized as the backbone.**

| Methods | CUB | Birds | Cifar100 | GTSRB | SVHN |
|---|---|---|---|---|---|
| Full [21] | 78.8 | 72.8 | 84.0 | 96.8 | 90.6 |
| VP [3] | 75.4 | 69.0 | 79.1 | 89.8 | 91.3 |
| VPT [23] | 72.1 | 65.3 | 72.8 | 88.5 | 61.8 |
| DAM-VP [20] | 79.7 | 71.4 | 81.8 | **92.8** | 89.3 |
| E$^2$VPT [16] | 73.3 | 66.8 | 80.4 | 89.2 | 80.3 |
| **InsVP** | **80.2** | **72.8** | **83.5** | 95.6 | **92.7** |

## 4.4 Comparison with State-of-the-arts

*4.4.1 Comparison on Pretrained ViT.* We first conduct experiments on ten popular datasets using the ImageNet-21k supervised ViT-B/16 [13] as the pretrained model. As shown in Table 1, compared with other SOTA parameter-efficient finetuning and visual prompting methods, our InsVP exhibits a notable improvement of **3.5**% and **6.0**% on the GTSRB and SVHN respectively. Furthermore, across the other eight datasets, InsVP all achieves the best performance. Overall, compared with the second-best player, the cluster-level prompting method DAM-VP, our InsVP achieves an average improvement of 2.8% across the ten datasets. This is because our InsVP leverages image-level and feature-level instance visual prompting to elaborately capture discriminative characteristics of individual

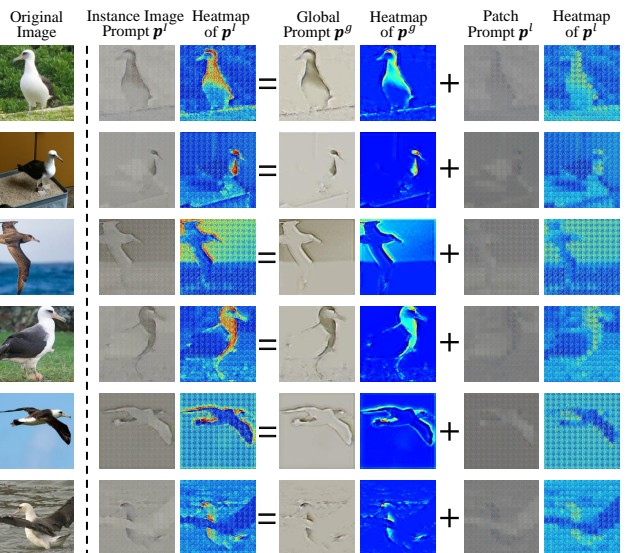

**Figure 4: Visualization of the generated instance image prompt $p^I$, global prompt $p^g$, and patch prompt $p^l$ through our InsVP method. The image prompt $p^I$ is derived by adding the global prompt $p^g$ and the patch prompt $p^l$ together.**

instances which enhance the pretrained model's recognition capability, leading to more accurate prediction results.

To verify this, we further present the obtained prompts and Grad-CAM visualization results of the samples in CUB. As shown in Figure 3 and Figure 4, the visualization results reveal that DAM-VP employs the exact same prompt for the samples belonging to the same category but exhibiting significant differences. Moreover, it seems that the learned prompts of DAM-VP lack a direct connection with the image samples or categories, and explicit semantic information is not observed. The Grad-CAM visualization results also indicate that under their prompts, the pretrained model fails to focus on discriminative regions of the instances. In contrast, our InsVP precisely outlines the object and identifies discriminative regions, such as the location of the bird's head and neck. Consequently, the pretrained model can more precisely focus on the object itself rather than the background, leading to outstanding performance.

*4.4.2 Experiments on VTAB-1k Benchmark.* To further validate the effectiveness of our InsVP, in addition to the ten datasets mentioned above, following [16, 23], we also conduct experiments on another widely used VTAB-1k [52] benchmark. As shown in Table 2, compared to the second-best method, E$^2$VPT [16], InsVP achieve improvements of 1.8%, 0.8%, and 1.0% in the three different tasks *Natural*, *Specialized*, and *Structured*, respectively. This further illustrates the robust adaptability of our instance-level visual prompting designed based on the original image across diverse tasks.

*4.4.3 Comparison on Different Model Architectures.* To verify the generalization ability of our InsVP across different model architectures, we conduct experiments using the Swin Transformer [30] as the backbone. As shown in Table 3, although the Swin Transformer

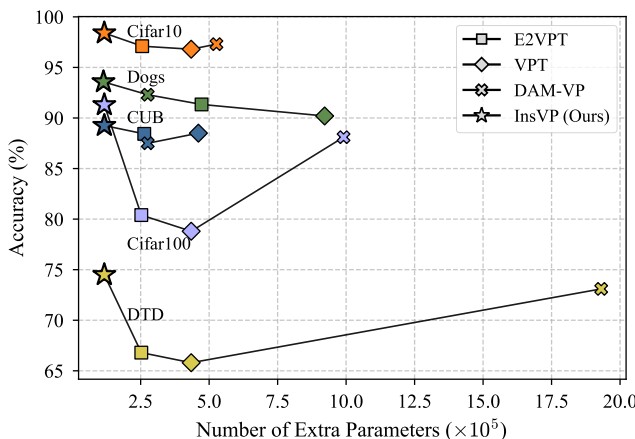

Figure 5: The comparison results of InsVP with other visual prompting methods in the number of extra parameters and model performance.

Table 5: Ablation study about the influence of components in InsVP. "-" and "✓" represent without or with this component. $p^I$ represents the instance image prompt, comprising patch prompt $p^l$ and global prompt $p^g$, and $p^F$ represents the instance feature prompt, consisting of common prompt $p^c$ and specific prompt $p^s$.

| $p^I$ | | $p^F$ | | CUB | Birds | Cifar100 | SVHN |
|---|---|---|---|---|---|---|---|
| $p^l$ | $p^g$ | $p^c$ | $p^s$ | | | | |
| - | - | - | - | 85.3 | 75.9 | 63.4 | 36.6 |
| ✓ | - | - | - | 87.1 | 83.1 | 85.5 | 85.3 |
| ✓ | ✓ | - | - | 87.9 | 83.5 | 87.0 | 91.8 |
| ✓ | ✓ | ✓ | - | 88.9 | 84.3 | 90.3 | 94.5 |
| ✓ | ✓ | ✓ | ✓ | **89.3** | **84.6** | **91.2** | **96.1** |

is a more advanced model that utilizes the shifted windows, compared with other visual prompting methods, our InsVP exhibits a consistent improvement of 1% to 3% across five datasets. This is because our InsVP is not designed for a specific network architecture. Instead, our InsVP directly extracts discriminative characteristics from each instance image itself, making it compatible with various pretrained model architectures.

*4.4.4 Comparison on Different Pretraining Methods.* In addition to supervised training, self-supervised contrastive learning, such as the MoCo paradigm [10], is also a commonly used method for model pretraining. To validate the generalization of our InsVP across different pretraining techniques, we conduct experiments based on the pretrained ViT-B/16 backbone via the MoCo v3 paradigm [10]. As reported in Table 4, our InsVP outperforms other visual prompting methods with an improvement ranging from 1% to 3% across five datasets consistently. This superiority is attributed to that our approach is directly tied to the characteristics of the data, showcasing enhanced adaptability to different pretraining paradigms compared to other methods.

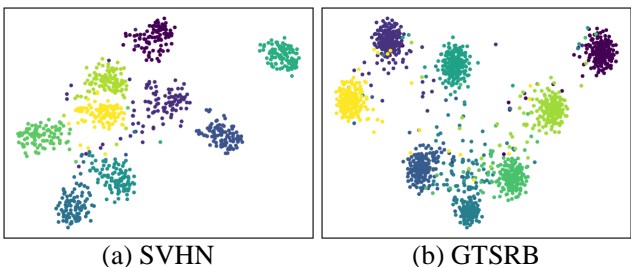

Figure 6: The t-SNE visualization results of specific prompt $p^s$ generated by specific prompter $\mathcal{G}_s$.

*4.4.5 Comparison of Extra Parameter Overhead.* For visual prompting methods, the number of introduced extra parameters is a crucial factor in determining their practical applicability. As illustrated in Figure 5, we compare the additional parameter quantities introduced by InsVP and other visual prompting methods. In the case of task-level visual prompting methods, since they use a shared prompt for all samples, a significant number of learnable tokens are required to capture diverse features of different samples. On the other hand, for cluster-level visual prompting methods like DAM-VP, a set of prompts needs to be learned for each cluster, resulting in a substantial increase of extra parameter overhead, reaching several times or even hundreds of times [20], thereby compromising the scalability.

In contrast, our proposed InsVP adopts a more straightforward methodology by extracting crucial prompting information directly from raw images. This allows InsVP to efficiently capture discriminative characteristics by lightweight prompters. Therefore, as shown in Figure 5, InsVP achieves optimal performance with minimal parameter costs.

## 4.5 Ablation

*4.5.1 Influence of Different Components.* To verify the effectiveness of different prompts in our proposed InsVP, ablation experiments are conducted on four datasets and reported in Table 5. As demonstrated, when neither component is used, the InsVP is degraded to a frozen pretrained ViT model with a learnable classifier. Taking results on CUB as an example, when using only the patch prompt $p^l$, the model's performance improved by 1.8%. Furthermore, when both the patch prompt $p^l$ and the global prompt $p^g$ are used simultaneously, the model's performance further increases by 0.8%. This is because the local information captured by the patch prompt $p^l$ and the global information in the global prompt $p^g$ can complement each other, allowing for more accurate extraction of discriminative information of individual instances. Moreover, when adding the feature-level common prompt $p^c$, it captures the common characteristics of all instances, resulting in an additional 1.0% improvement in model performance. Finally, with the addition of the generated specific feature prompt $p^s$, the complete InsVP is obtained. The specific prompt $p^s$ extracts unique discriminative characteristics from individual instances, enabling the model to more accurately extract features for each instance and achieve the

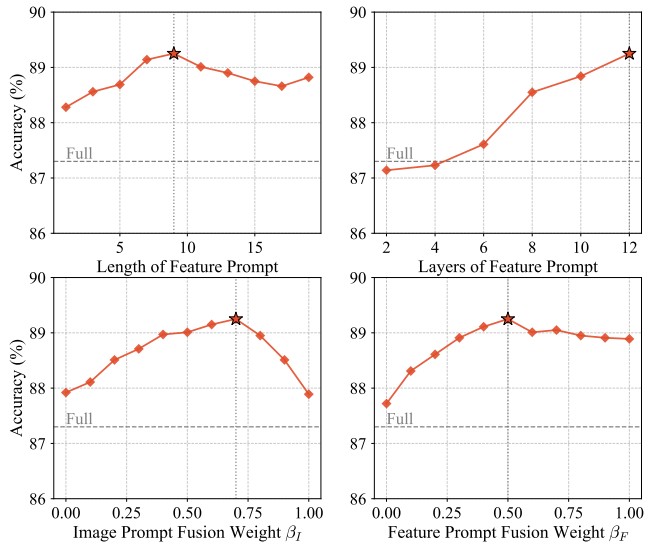

Figure 7: Influence of hyper-parameters of InsVP in CUB.

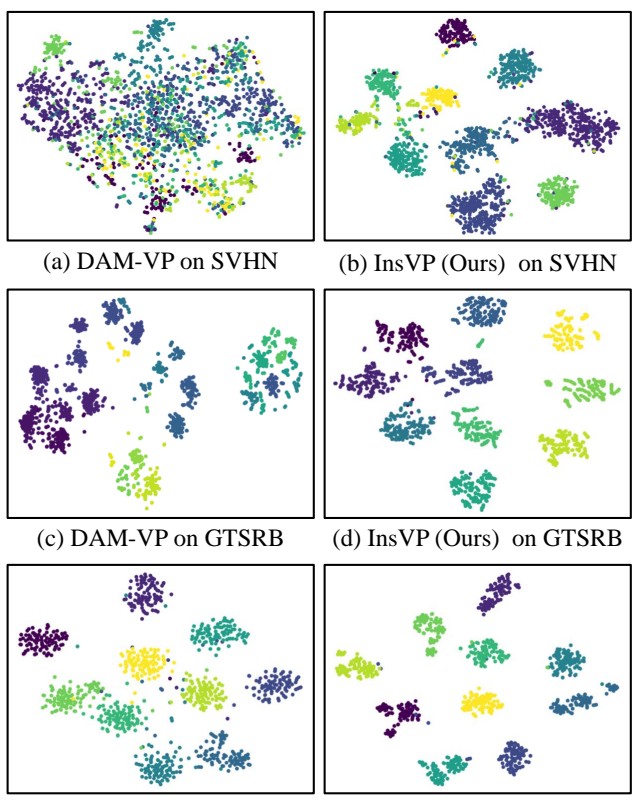

(a) DAM-VP on SVHN     (b) InsVP (Ours) on SVHN

(c) DAM-VP on GTSRB     (d) InsVP (Ours) on GTSRB

(e) DAM-VP on Cifar100     (f) InsVP (Ours) on Cifar100

Figure 8: Feature t-SNE visualization results for InsVP and comparison method DAM-VP on three datasets.

best performance. The results on other datasets also demonstrate a consistent trend to that observed on CUB.

*4.5.2 The t-SNE Visualization Results of Specific Prompt $p^s$.* To further explore the impact of the specific prompt, we perform t-SNE visualization for specific prompt $p^s$ generated by specific prompter $\mathcal{G}_s$ on SVHN and GTSRB datasets. As illustrated in Figure 6 below shows a notable correlation between specific prompts' distribution and sample categories. Despite being added to the middle layer of the network, specific prompts effectively capture discriminative information unique to individual instances, showing variability across different categories.

*4.5.3 Influence of Hyper-parameters.* The length of the feature prompt $L_p$ and the number of MSA layers applied are two crucial hyper-parameters in our InsVP. To investigate their impact, we have conducted extensive ablation experiments. As depicted in Figure 7, the model's performance initially improves and then declines with the gradual increase of the feature prompt's length. This behavior arises due to the overfitting caused by an excessively large number of parameters with limited training data. Regarding the experiments on MSA layers, our feature prompt attains the best results when applied across all layers of ViT. This is because, at this point, the prompt can adapt the pretrained model across all network layers, enabling the pretrained model to better accommodate downstream tasks and consequently achieve superior performance.

We also conduct ablation experiments to assess the impact of image prompt fusion weight $\beta_I$ and feature prompt fusion weight $\beta_F$. As depicted in Figure 7, when these two hyper-parameters are either too large or too small, the performance is degraded. When choosing intermediate values, patch prompt $p^l$, global prompt $p^g$, specific prompt $p^s$, and common prompt $p^c$ can readily complement each other, fully unleashing the potential of our method.

*4.5.4 The t-SNE Visualization Results of Extracted Features.* As shown in Figure 8, we visualize the features obtained by InsVP and

DAM-VP via t-SNE [42]. From the visualization results, it is evident that the features extracted by DAM-VP from samples of the same category are relatively scattered, and some are mixed with features from other categories. In contrast, the features extracted by our InsVP from the same category are tightly clustered together, and they exhibit good distinctiveness from features of other categories. This is attributed to our instance image prompt and instance feature prompt, which can recognize the most essential discriminative characteristics of different category samples.

## 5 CONCLUSION

In this paper, we propose a novel and efficient instance visual prompting method, named InsVP. In comparison to the task-level or cluster-level visual prompting methods, our instance-level InsVP achieves outstanding performance by extracting discriminative characteristics of the individual instances using the proposed instance image prompt and instance feature prompt. We demonstrate that generating prompts directly from the original image itself is more efficient and the visualization results also illustrate our prompts are closely related to individual instances. In the future, it is interesting to further investigate how to explicitly explore hierarchical instance prompting to tackle instances with various recognition difficulties.

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
