# OpenReview forum: "InsVP: Efficient Instance Visual Prompting from Image Itself"
_acmmm.org/ACMMM/2024/Conference — MM2024 Poster_

### Official Review · Reviewer_rhUs · 2024-05-16

**Rating:** 6
**Confidence:** 3

**Summary:**

The paper propose an Instance Visual Prompting method to capture nuanced differences among instances with a few parameters. This method leverages the discriminative instance-specific information of the input image itself to enhance the recognition capability of pretrained models.

**Strengths:**

This paper proves the effectiveness of each module through a large number of experiments and visualizations, and the experiments are sufficient and persuasive. Full text writing expression is smooth, clear and easy to understand.The analysis results are reasonable and compared with other methods in detail.The method presented in this paper achieves satisfactory result.

**Limitations:**

It would be better if the article could analyze whether it can be applied to other fields and practical application scenarios.

**Suitability:**

3

---

### Official Review · Reviewer_T8nJ · 2024-05-23

**Rating:** 3
**Confidence:** 4

**Summary:**

The paper proposes InsVP for efficiently fine-tuning the pre-trained vision models. The instance image prompt extracts discriminative information from the original image. Experimental results show that InsVP outperforms existing methods with fewer parameters.

**Strengths:**

- This paper is well-structured.
- Extensive experiments are conducted.
- The proposed method requires fewer parameters and outperforms other SOTA methods.

**Limitations:**

- Lack of novelty. The authors do not provide a comprehensive literature review to establish its novelty. They claim to propose a novel instance visual prompting method, but the concept of visual prompting from the image itself has already been explored in previous works[1, 2, 3]. They must be discussed. Quantitative analysis would be better.

- Lack of efficiency analysis. Despite certain improvements, a detailed efficiency analysis is recommended.


[1] Prompt Generation Networks for Input-based Adaptation of Frozen Vision Transformers.

[2] Explicit Visual Prompting for Low-level Structure Segmentations.

[3] LION: Implicit Vision Prompt Tuning.

**Suitability:**

1

---

### Official Review · Reviewer_pAV2 · 2024-05-24

**Rating:** 5
**Confidence:** 3

**Summary:**

This paper introduces the Instance Visual Prompting (InsVP) method, which significantly enhances the finetuning of pretrained visual models by capturing both commonalities and unique discriminative characteristics of individual instances. Unlike existing methods, InsVP employs instance-specific image and feature prompts, leading to superior performance and efficiency. Extensive experiments demonstrate that InsVP outperforms state-of-the-art methods with fewer parameters, and the authors plan to release the code for further research and application.

**Strengths:**

1.The paper reads well and its easy to follow.
2.The conept of  Instance Visual Prompting (InsVP) is interesting.

**Limitations:**

1.The customized prompts might require significant training resources compared to current methods. Additionally, there is a concern about the generalizability of this instance-specific approach in base-to-novel tasks.

2.The feature layer prompt appears similar to the method used in maple[1]. Please explain the differences between your approach and maple.
[1] MaPLe: Multi-modal Prompt Learning
3. It is recommended to include comparative experiments with maple to better illustrate the advantages of your method.

**Suitability:**

3

---

### Official Review · Reviewer_dV7A · 2024-05-29

**Rating:** 3
**Confidence:** 4

**Summary:**

InsVP (Instance Visual Prompting) utilizes the discriminative features of individual instances within input images to enhance the recognition capabilities of pre-trained models. The methodology in this paper includes two internal modules: Image-level Instance Visual Prompting and Feature-level Instance Visual Prompting. The former extracts crucial and subtle discriminative information directly from the raw images, imposing this information onto the input images to highlight the distinctions between different instances. The latter captures the commonalities and characteristics among individual instances, inserting this information into the model's intermediate layers, thereby bolstering the pre-trained model's ability to discern the unique features of each instance. Through these two modules, InsVP achieves more precise prediction on down-stream tasks.

**Strengths:**

1. InsVP provides more refined visual cues, thereby improving model performance.
2. The writing is good, and the figures are well-drawn.

**Limitations:**

1. The Image-level Instance Prompting module appears more as an image enhancement method rather than a visual prompting method for updating based on frozen networks, seeming to depart from the definition of visual prompt tuning.
2. The ablation studies are insufficient. The effects of using the Feature-level Prompt method alone or comparing it to patch or global prompts have not been explored. It makes me confused whether the improvements are mainly due to the image enhancement module, i.e., the Image-level Instance Prompting module.
3. Table 1 includes comparisons among methods employing various backbones. However, it's unclear whether comparing across different backbones is fair. Additionally, it's not specified which backbone is used for the plug-and-play methods.
4. The proposed method introduces additional parameters due to the incorporation of several extra modules, such as the global prompter, local prompter, and specific prompter. However, Figure 5 demonstrates that the proposed method has fewer parameters compared to VPT. The authors should clarify why their method results in fewer parameters despite the added modules. Additionally, a comparison of the inference burden is necessary.
5. Figure 3/4 have shown some intermediate results (i.e., Grad-CAM / t-SNE figures) of the proposed method compared with DAM-VP method. Why you choose DAM_VP method for such kind of comparison? Can you choose more latest methods for further comparison and analysis?

**Suitability:**

3

---

### Meta-Review · Area_Chair_tHKP · 2024-07-07

**Recommendation:** Accept (Poster)
**Confidence:** 4

**Metareview:**

This paper proposes a novel visual prompting method that uses each image (and its feature) as the corresponding prompt, therefore improving the adaptation to down-stream tasks. Initially the paper got 1 borderline accept, 1 weak accept and 2 borderline reject. Reviewers raised questions about the novelty, the additional training cost, the ablation study, etc. The authors provided responses to these questions point to point. Two reviewers are satisfied with the responses and maintained positive ratings. Reviewer pAV2 still had concern on the training cost and maintained BR. The AC thinks the idea instance prompting is novel  and the performance improvement is valuable, and thus recommends accepting this paper.